# Sickle Cell Disease: A Paradigm for Venous Thrombosis Pathophysiology

**DOI:** 10.3390/ijms21155279

**Published:** 2020-07-25

**Authors:** Maria A. Lizarralde-Iragorri, Arun S. Shet

**Affiliations:** Laboratory of Sickle Thrombosis and Vascular Biology, Sickle Cell Branch, National Institutes of Health, Bethesda, MD 20892, USA; maria.lizarralde-iragorri@nih.gov

**Keywords:** venous thromboembolism (VTE), sickle cell disease (SCD), mice models, stasis, hypercoagulability, endothelial injury, inflammation

## Abstract

Venous thromboembolism (VTE) is an important cause of vascular morbidity and mortality. Many risk factors have been identified for venous thrombosis that lead to alterations in blood flow, activate the vascular endothelium, and increase the propensity for blood coagulation. However, the precise molecular and cellular mechanisms that cause blood clots in the venous vasculature have not been fully elucidated. Patients with sickle cell disease (SCD) demonstrate all the risk factors for venous stasis, activated endothelium, and blood hypercoagulability, making them particularly vulnerable to VTE. In this review, we will discuss how mouse models have elucidated the complex vascular pathobiology of SCD. We review the dysregulated pathways of inflammation and coagulation in SCD and how the resultant hypercoagulable state can potentiate thrombosis through down-regulation of vascular anticoagulants. Studies of VTE pathogenesis using SCD mouse models may provide insight into the intersection between the cellular and molecular processes involving inflammation and coagulation and help to identify novel mechanistic pathways.

## 1. Venous Thromboembolism and Sickle Cell Disease

The third most common cause of vascular death from thrombosis in the United States is venous thromboembolism (VTE), a disorder that includes both deep vein thrombosis (DVT) and pulmonary embolism (PE) [1]. Many factors influence VTE incidence including increasing age, obesity, hospitalization, active cancer, immobility, pregnancy, estrogen therapy, and the presence of underlying inflammatory disease [2]. Our overall understanding of the molecular pathogenesis of VTE has been largely advanced by development of animal models that have been engineered to recapitulate human disease [3]. From these studies, it is apparent that activation of coagulation and inflammatory pathways are critical to VTE pathophysiology. However, the precise molecular and cellular events leading to the initiation, development, and resolution of VTE in humans is yet to be completely elucidated. In this review, we explore how the pro-inflammatory and procoagulant milieu in sickle cell disease (SCD) predisposes to VTE, and review whether elucidating these intersecting pathways may advance our understanding of the pathobiology of venous thrombosis.

SCD is the most prevalent genetic disorder worldwide, with estimates suggesting that approximately 300,000 infants are born every year with this condition [4]. It is a multisystemic disorder caused by a single point mutation in the sixth codon (c.20A > T) of the hemoglobin gene, leading to substitution of valine for glutamic acid in the β-globin chain [5]. Sickle hemoglobin (HbS) polymerizes when deoxygenated, forming insoluble fibers that cause “sickled” red cells and trigger a complex cascade of events leading to protean manifestations of SCD [6,7,8].

Hemolytic anemia and vaso-occlusive pain crises (VOC) are dominant clinical manifestations of SCD, but patients are also at risk for developing VTE. In support of this notion are observational studies, indicating a 25% increased VTE risk with a mean age of incidence very similar to high-risk thrombophilia patients (30 vs. 29 yrs.), which sharply contrasts with age of incidence in the general population (65 yrs.) [9,10,11,12]. Moreover, adults with wide range of sickling hemoglobinopathies have been associated with the development of VTE, from hemoglobin SS to compound heterozygous states to sickle cell trait [13]. A curious phenomenon in SCD patients is the increased overall prevalence of pulmonary embolism (PE) that occurs in the absence of detectable extremity deep venous thrombosis (DVT) suggesting either the occurrence of in situ pulmonary thrombosis [14] or classic thromboembolic PE from clot friability/instability [15]. In SCD, thrombosis also affects the arterial circulation, leading to devastating complications, such as stroke and silent cerebral infarction [16], a topic that is beyond the scope of this review.

According to Virchow, the occurrence of venous thrombosis can be predicted by a triad of blood hypercoagulability, changes in the vessel wall, and stasis [17]. All the components of Virchow’s triad, i.e., hemostatic features indicative of hypercoagulable blood, cellular biology characteristic of endothelial activation/inflammation, and rheological features consistent with vascular stasis, are demonstrable in SCD patients. It is therefore unsurprising that venous and arterial thromboses are frequently documented in patients with SCD [18]. However, the exact sequence of events leading to venous thrombus formation is less clear, as is the relative contribution of blood cells/vessel wall and blood flow/stasis. Blood coagulation and innate immune responses are closely interrelated, thus the presence of dysregulation of inflammatory and coagulation pathways in SCD suggests that they contribute to VTE pathophysiology. Key amongst these dysregulated processes is the overexpression of tissue factor (TF), the principal trigger of human coagulation in vivo, and a coagulant protein critical to atherothrombosis [19]. The generation of thrombin in SCD occurs via initiation of coagulation by the TF/VIIa pathway and the contact pathway [20,21]. Sickle blood also exhibits increased TF-dependent procoagulant activity, and consequently increased thrombin generation, as reflected by elevated levels of thrombin anti-thrombin complexes (TAT), prothrombin fragment 1.2 (F1.2), and D-dimers [22,23,24,25]. Since the molecular and cellular events involved in vascular thrombosis are complex and involve many overlapping pathways, the use of murine models could provide important insight into the processes, particularly those dysregulated inflammatory and coagulation pathways in SCD [26] that are critical to VTE pathogenesis.

## 2. Animal Models of Disease Pathophysiology

Elucidating the molecular and cellular events leading to vascular thrombogenesis has largely been advanced by the use of various different animal models but for the purpose of this review, we focus only on studies of murine models. Murine models have distinct advantages, and it is worth noting that, among the murine models in vogue, no single model unequivocally replicates human venous thrombosis pathophysiology. Similarly, murine models of SCD have elucidated the multistep mechanistic processes thought to underlie the development of acute vaso-occlusive pain crisis (VOC) in patients with SCD. Again, whether these murine vascular events that are induced by infusion of cytokines or heme truly represent spontaneous VOC in humans is largely unknown. Regardless of these considerations, murine models offer unique advantages, such as low maintenance cost, a short life span, rapid reproduction rate, and ease of genetic manipulation, explaining their widespread use in clinical investigation.

### 2.1. Murine Models of VTE 

Excepting in surgery associated DVT, histopathological examination of human vein thrombosis in the region of the clot seldom indicates evidence for vein injury. Thus, human DVT differs from DVT in animal models, where injury of the vein, even if only by ligation, is usually an initiating event. From this perspective, describing the various methodologies utilized to develop vascular models that replicate human disease pathophysiology provides important contextual information. Surgical exposure of a target vein and provoking venous thrombosis by inducing endothelial inflammation/injury is an important limitation of most mouse models. A major concern here is that the nature and extent of vascular injury is variable, and greatly influences subendothelial TF exposure. In most murine VTE models, the inferior vena cava (IVC) is used to induce venous thrombosis, due to its size and easy accessibility. The IVC stenosis approach possibly represents human pathophysiology more accurately by minimizing endothelial injury typically associated with previous models.

Murine models incorporating heritable thrombophilia mutations have yielded insight into the pathophysiology of VTE induced by abnormal blood coagulation [27]. Unfortunately, many of these studies fail to capture the effects of prolonged exposure of the vascular endothelium to thrombophilia that possibly contribute to chronic venous vasculopathy [28]. Blood stasis is another predictor of VTE in humans, as noted by the increased propensity for VTE in individuals with prolonged immobility. VTE models recapitulate stasis-induced VTE by the use of a surgical ligature to reduce the IVC lumen by 80–90%. Reduction of the vessel lumen does not denude the endothelium; however, the endothelium is still activated, and releases the von Willebrand factor (vWF) and P-selectin from the Weibel-Palade bodies. Information regarding thrombus size and composition, thrombus resolution and eventual vessel recanalization is forthcoming, but the lack of valves in the IVC is a distinct disadvantage to studying DVT [3,29].

Transgenesis and gene targeting techniques to make an organism’s genes inoperative (knock out) or mutate them or replace them with genes from another species have led to an understanding of the events occurring during vascular thrombosis and fibrinolysis [28]. Since VTE is influenced by heritable or acquired thrombophilia genes, inducing these genetic mutations in mice may be used to study the direct effects of these gene mutations on VTE development. Depending on the technique used to induce thrombogenesis, one can study thrombus formation resulting from: (1) venous stasis, (2) endothelial injury, or (3) induced localized hypercoagulability (Table 1).

### 2.2. Sickle Cell Disease Animal Models

Transgenic mice expressing βS-hemoglobin have enhanced our understanding of different clinical aspects of SCD, such as anemia, vaso-occlusion, and chronic organ damage [30,31]. In particular, subjecting mice to hypoxia followed by reoxygenation or infusion of hemoglobin appears to successfully mimic VOC in human SCD, and have provided insight into vascular pathobiology SCD [32,33,34,35]. However, the choice of murine model can influence study results and outcome. For instance, β^SAD^ mice tolerate hypoxia induced events relatively well and Aβ^S Antilles^ mice can be surgically manipulated to study vaso-occlusion [34]. The more severe humanized models do not tolerate either surgical manipulation or hypoxia. Moreover, the mouse background and basal murine hemoglobin expression can interfere with HbS polymerization and mask the true phenotype of sickling hemoglobinopathies [30,36].

Humanized transgenic mice models have, at least to a certain extent, overcome pathophysiological limitations, permitting the conduct of mechanistic studies. The Townes and Berkeley(BERK) sickle mouse models show characteristic features of SCD, including severe anemia, hemolysis, inflammation, and endothelial activation [36,37]. Importantly, they also display biochemical features consistent with the sickle hypercoagulable state (see Table 2), e.g., increased thrombin generation [38,39], depletion of the natural anticoagulants [28,38], increased TF expression [40], and abnormalities in fibrinolytic activity [15]. Evidence of thrombosis can also be found in multiple organs of sickle cell mice [41], and hypoxia further enhances thrombosis in their pulmonary vasculature [42]. Insights into molecular and cellular events occurring at the fluid-wall interface of blood vessels that are critical to both VTE and SCD pathophysiology have been gleaned from several of these models, as discussed further below.

## 3. Insights into VTE Pathophysiology Using SCD Mouse Models

Since all three components of Virchow’s triad are encountered in SCD (Figure 1), employing sickle cell mouse models to enhance our understanding of VTE pathobiology appears to be specifically advantageous. From this perspective, reviewing the dysregulated coagulation and inflammation pathways that probably facilitate VTE development in patients with SCD is relevant.

The root cause of sickle cell pathology is the polymerization of HbS, a process influenced among other factors largely by allosteric effectors and delayed transit time of sickled red blood cells (RBCs) in the microcirculation. Consequently, the sickle RBC becomes fragile, less deformable, and prone to lysis, releasing proinflammatory hemoglobin and extracellular vesicles (EVs). Moreover, frequent episodes of vaso-occlusive crises in patients lead to ischemia, followed by reperfusion, which is profoundly inflammatory to blood and vascular endothelial cells. The molecular, cellular and plasmatic events occurring might be represented by Virchow’s triad that summarizes factors predisposing to pathological venous thrombosis. Although the exact sequence of events is unclear, it appears that activation of coagulation by TF is an important trigger of clinical thrombosis. Endothelial cell (EC) injury, as a consequence of plasma cell free-heme induces abnormal surface expression of tissue factor (TF), P-selectin, von Willebrand Factor (vWF). Moreover, reactive oxidative species (ROS) production, lowered vascular nitric oxide (NO), and endothelial/leucocyte activation leads to release of other damage-associated molecular patterns (DAMPs), EVs, and formation of aggregates that support procoagulant and proadhesive vascular phenomena. Several other vascular factors altering HbS polymerization and red cell delay time impair microvascular blood flow, causing venous stasis and augmenting prothrombotic phenomena. Taken together, these events collectively initiate and propagate venous thrombosis and lead to clinical deep vein thrombosis or pulmonary embolism in SCD. Figure created with BioRender.com.

### 3.1. Hypercoagulability

A hypercoagulable state or hypercoagulability describes a condition where blood clots spontaneously with little or no provocation. Pathological hypercoagulability might occur, due to increased procoagulant proteins, the presence of clotting protein variants that are more procoagulant, decreased anticoagulant proteins, and/or decreased fibrinolysis. The mechanism by which intravascular venous thrombosis occurs is not fully elucidated, but is generally agreed to involve TF, as the initiator of pathological coagulation [43]. Several models in which “blood borne” TF, probably associated with blood cells or EV, provide support for this notion [44,45]. In these studies, depending on the type of vessel injury induced and the vascular bed involved, blood borne TF contributed little to stasis induced venous thrombosis, indicating that TF in the vascular clot was derived from the vessel wall [46]. This evidence bolsters support for the importance of endothelial TF expression and pathological thrombosis in SCD. Physiological processes such as shear stress, inflammation, and hypoxia can promote vascular luminal TF expression, and subsequent fibrin deposition in mouse models [47,48]. Moreover, in vivo and in vitro evidence of the pro-inflammatory effects of intravascular heme in SCD demonstrate its capacity to elicit pathological TF expression in SCD patients and murine models (see Section 3.2 below) [25,40,49,50,51]. Evidence for cell and extracellular vesicle (EV) associated-TF, in patients at baseline and during VOC [25], indicate the pathophysiological role that EV-associated TF plays in murine models of VTE [52,53,54,55]. Surface TF expression on vascular endothelial and blood cells is associated with procoagulant effects and proinflammatory signaling [56]. In murine SCD models, TF originating from hematopoietic cells and vascular endothelial cells appear to have distinct effects on systemic inflammation and hypercoagulability [39,57,58]. Collectively, these studies implicate TF as a critical mediator of vascular inflammation and thrombotic end-organ damage in murine models of SCD and suggest that appropriately targeted anticoagulation treatment might lead to improved clinical outcomes in sickle cell patients.

Unlike the TF pathway, the contact pathway lacks a physiological role in hemostasis, but it appears to be involved in pathological venous thrombosis. In support of this idea is the observation that FXII deficiency in mice led to attenuation of thrombosis, and conferred a survival advantage following the induction of lethal pulmonary embolism [59]. FXI deficiency and inhibition of FXII-dependent FXI activation does not impact thrombin generation in sickle cell mice, but FXII deficiency attenuates thrombin generation in sickle mice [60]. Cell death during I/R injury [61] and release of histone-DNA complexes (nucleosomes) initiate inflammation, and can propagate thrombosis and lead to fibrin deposition [62,63]. In SCD patients, the presence of the following substances in blood, e.g., negatively charged cell surface membranes phosphatidylserine (PS) exposure on red and other cell-derived EVs, nucleic acids [64], and platelet derived polyphosphates, explain contact pathway activation [18,21,65,66]. Moreover, the presence of these activating substances during acute complications and the subsequent low-grade thrombin generation suggests the involvement of this pathway in VOC pathophysiology. Regardless, at present there is no evidence for a direct effect of contact pathway activation on VTE development in both humans and mouse models of SCD.

The activation of both TF and contact pathways leads to the accumulation of intravascular thrombin, a potent thromboinflammatory signal. Murine coagulation models have delineated the critical role of thrombin in the vascular pathobiology of VTE [67]. In BERK SCD mice, elegant studies that attenuated thrombin either via genetic manipulation or antisense oligonucleotides have demonstrated its role in sickle vasculopathy. Diminished thrombin markedly reduced plasma inflammatory markers, D-dimers and endothelial cell dysfunction [68]. Moreover, histopathological analysis performed on at least 12-month aged prothrombin deficient mice revealed attenuation of chronic vasculopathy and target organ damage, compared to SCD mice with normal prothrombin levels. Prolonged exposure of the vasculature to thrombin seemingly led to organ vasculopathy, that was likely mediated via vascular endothelial protease activated receptors (PARs). Interestingly, inhibiting PAR-1 either pharmacologically or using PAR-1 non-hematopoietic cell-deficient mice, significantly attenuated heme-induced microvascular stasis in a sickle mouse model of VOC [69]. TF inhibition also led to similar findings in the pulmonary microvasculature, suggesting that coagulation may play a role in microvascular stasis.

As described below, sickle hemolysis can consume vascular nitric oxide (NO) and plasma arginine, inducing profound endothelial and vascular dysfunction [70,71]. In particular, the endothelial hemostatic balance shifts to a thrombotic phenotype, with the upregulation of procoagulant molecules and the downregulation of anticoagulant molecules. Unsurprisingly, several studies have reported significant reduction of the anticoagulant factors, protein C and S levels in both children and adults with SCD in steady state, compared with controls [72,73]. Lowered protein C activity levels in SCD patients that decrease further during VOC, suggests that protein C may be consumed acutely and chronically. Another key protein of the protein C pathway is thrombomodulin (TM), an endothelial-bound protein which activates protein C [74]. At baseline, SCD mice exhibit elevated TM and endothelial protein C receptor (EPCR) expression, suggesting a compensatory up-regulation of these proteins in SCD [41]. While these data may seem preliminary, they provide evidence that acquired defects in anticoagulant proteins play a role in the multi-causal pathway of VTE pathogenesis in SCD.

### 3.2. Inflammation and Endothelial Injury

As alluded to before (see Section 3.1 above), healthy endothelium expresses anticoagulants, such as the TF pathway inhibitor (TFPI), TM, EPCR, and heparin-like proteoglycans, and ectonucleotidase CD39/NTPDase1, which metabolizes the platelet agonist ADP [75]. Moreover, the release of endothelial NO and prostacyclin is inhibitory to platelets. Several germline encoded pattern recognition receptors (PRR) e.g., the Toll-like receptors (TLR), the C-type lectin receptors, and the nucleotide binding domain-like receptors (NLR) mediate innate and protective immune responses [76]. In SCD, host-derived PRR activators [77] trigger these receptor pathways leading to sterile inflammation and immunothrombosis. Designated DAMP molecules (e.g., cell-free hemoglobin, high-mobility group box 1 (HMGB1) and extracellular purine nucleotides) play a fundamental role in VTE pathophysiology [77,78,79] and also seem to drive SCD pathophysiology [26,61,80,81]. Since the regulation of coagulation in non-inflammatory states is likely to be different from inflammation driven coagulation, understanding how venous thrombosis occurs as part of the inflammatory response is critical to achieving insight into the pathogenesis of hypercoagulability in SCD.

Hemolytic disorders characterized by intravascular hemolysis exhibit a variety of clinical symptoms including vascular thrombosis. Cell-free heme, an erythrocyte DAMP, induces a procoagulant endothelial phenotype due to surface expression of TF, vWF, and P-selectin and downregulation of TM [40,49,80,82,83,84,85]. Endothelial activation by heme occurs via TLR-4 signaling or by direct toxicity from erythrocyte derived EVs [82,86]. TLR-4-mediated activation of nuclear factor-κB transcription factor activity also upregulates endothelial cytokine generation (tumor necrosis factor-alpha (TNF-α) and interleukin1-beta (IL-1β), as well as TF expression [82,87]. Furthermore, heme induces assembly of the nucleotide-binding domain-like receptor protein 3 (NLRP3) inflammasome in endothelial cells and leukocytes, with the consequent secretion of IL-1β [26,77,88,89]. Finally, heme can activate complement via TLRs and P-selectin, leading to deposition of C3 on glomerular and hepatic endothelial cells and subsequent microvascular thrombosis [83,90]. Complement mediated vascular injury follows hemolytic crises and delayed hemolytic transfusion reactions in SCD patients [91], which, taken together with the observations in mice that complement directly activates platelets and TF to mediate venous thrombosis [92], suggests a role for investigating C5a targeted therapy. Although intravascular heme is rapidly bound by circulating haptoglobin and hemopexin in healthy individuals, these proteins are depleted in SCD patients [93] and replacement of either haptoglobin or hemopexin in SCD mice leads to a reduction in VOC [94]. The reticuloendothelial capacity for endocytosis and degradation of these heme binding protein moieties is also augmented by the upregulation of heme oxygenase 1 (HO-1). Interestingly, increased HO-1 activity in mice transplanted with sickle bone marrow appears to offer some protection from injury induced carotid artery thrombosis, although its protective effect against venous thrombosis is unknown [95].

High mobility group box 1 (HMGB1), a chromatin-binding protein responsible for maintaining DNA structure, is released by activated immune cells or necrotic tissues and plays a role in thrombosis by signaling through TLR4 [96]. Circulating HMGB1 derived from platelets recruits’ monocytes, primes the leukocyte inflammasome inducing NETosis and TF production, eventually leading to increased platelet aggregation and VTE [78]. Interestingly, plasma HMGB1 is increased in both humans and mice with SCD, and sickle patient plasma demonstrates increased HMGB1-dependent TLR4 activity compared with control plasma [89,97]. Activation of the neutrophil inflammasome by heme leads to NETosis and release of neutrophil extracellular traps (NETs) [98], an important immune response mechanism that has a role in VTE pathogenesis by acting as a scaffold for fibrin and assembly of coagulation complexes [99]. NETs activate platelets and enhance the recruitment of leucocytes via the platelet glycoprotein Ibα receptor, a process facilitating blood coagulation and thrombus propagation [100,101]. Platelet activation promotes adhesive interactions between SS RBCs and endothelial cells, and facilitates formation of aggregates with RBCs, monocytes and neutrophils [22,102,103,104,105,106], which, by hindering the microvascular blood flow, also augment stasis. Heterotypic aggregates are facilitated by protein-protein interactions between E-selectin and E-selectin ligand, which, in SCD mice, activates the leukocyte integrin αMβ2 [107].

Although platelet activation and alterations of platelet function in SCD are frequently described, the fundamental basis for these observations is less clear. Macro-thrombocytosis and platelet hyperresponsiveness to submaximal doses of thrombin in vitro have been observed in both humans [108] and SCD mouse models [109]. Both SCD patients and murine models also consistently demonstrate platelet NLRP3 inflammasome activity, suggesting an autocrine feedback loop for IL-1β, driven by the priming of innate immune and endothelial cells [88,89]. IL-1β derived from platelets is released mainly as extracellular vesicles through mechanisms dependent on NLRP3 activation that are triggered by mitochondrial ROS [88]. DAMPs, such as heme, can directly activate platelets by binding with the glycocalicin domain of GP1bα [110] or activate the platelet NLRP3 inflammasome [89]. Other DAMPs released during ischemia reperfusion injury, i.e., cell-free DNA and histone also activate platelets via their TLRs, leading to thrombin generation [111]. Given the evidence derived from randomized clinical studies showing a 30 to 50% risk reduction in VTE by primary or secondary thromboprophylaxis with ASA [112], it is reasonable to assume that platelet activation in SCD may play a role in VTE pathophysiology. Complement mediated platelet activation contributes to venous thrombosis in murine models, providing another link between innate immunity and coagulation [92]. Activated platelets form homotypic and heterotypic cell-aggregates, and platelet-neutrophil aggregates contribute to pulmonary arteriolar microemboli and stasis (see below) [103,105,106]. Autopsy evidence of dense platelet-rich thrombi in the pulmonary vasculature in over half of the SCD patients experiencing acute chest syndrome (ACS) [113], coupled with the findings of pulmonary arteriolar microthrombi in murine models of ACS [114], indicate that in situ thrombosis in the lung involves platelets. In addition, murine SCD models exhibit higher pulmonary vein TF expression, which is augmented by I/R [42,87], suggesting vulnerability of the pulmonary vasculature to in situ thrombosis [115]. These findings are directly in line with the clinical observation of in situ pulmonary thrombosis without detectable DVT in individuals with sickle cell disease [9,14,113] and trait [116]. Finally, thromboinflammation involves the platelet C-type lectin-like receptor 2 (CLEC-2), and in the IVC thrombosis model, CLEC-2 deficient mice appear to be protected from developing DVT [117]. Investigating a possible role for CLEC-2 and its ligand, podoplanin, using SCD mice may identify a novel therapeutic target in SCD.

### 3.3. Blood Stasis

Sickle red cells demonstrate increased endothelial adherence leading to stasis, which in combination with hypoxia promotes sickling in the venous circulation and predisposes to VOC. Murine models provide specific evidence for sluggish blood flow and “logjamming” in the post capillary venules [118]. Endothelial surface expression of cell adhesion molecules (intercellular adhesion molecule (ICAM) and vascular cell adhesion protein (VCAM) and selectins (E and P-selectin) is important for VOC and VTE pathophysiology [33,50,119]. Abnormal cell surface P-selectin expression on endothelial cells and platelets in SCD, facilitates homo/heterotypic cell-cell interactions and aggregate formation, further worsening stasis and vaso-occlusion [114,120,121,122,123]. Interestingly, stasis-induced IVC thrombosis does not develop in P-selectin-deficient mice, due to a failure in leukocyte recruitment and diminished leukocyte-vessel wall endothelial interactions [123]. The clinical relevance of these observations is supported by the findings of reduced VOC events in SCD patients, following treatment with crizanlizumab, a monoclonal antibody against P-selectin [124]. Several alternate P-selectin inhibitors are undergoing evaluation in animal models or humans, to either modulate or treat SCD [125]. Studies of these inhibitors in SCD may be indicated to see whether there is diminished stasis and lowered VTE frequency.

Under healthy conditions, the valve pocket endothelium has a thromboresistant phenotype consisting of increased EPCR and TM expression, and reduced vWF, which helps to prevent thrombosis [126]. Otherwise, vorticial blood flow, low oxygen tension, and stasis would lead to frequent valve pocket sinus thrombosis. Increased P-selectin and TF expression in response to hypoxia, possibly explains higher expression of these molecules on endothelial vein valve sinus pockets, and why these are usually the sites where thrombosis occurs [3]. Similarly, endothelial inflammation in SCD leads to increased TF and P-selectin expression and release of vWF that could promote small thrombi within the valve pocket, which, growing slowly over days to weeks, may eventually completely occlude the vessel. Interestingly, the venous thrombus in an IVC stasis model in SCD mice demonstrated ultrastructural differences with higher amounts of fibrinous material and red cell entrapment with extensive sickling [15]. In ex vivo sickle whole blood clots, the number of RBCs extruded from the clot was also significantly reduced, compared with the number released from sickle cell trait and non-sickle clots in both mice and humans. In addition, whole blood sickle clots were resistant to fibrinolysis by tissue plasminogen activator and RBC exchange transiently reversed this, in part by decreasing platelet-derived PAI-1. There is a growing appreciation of the role of red cells in thrombosis, in particular, how they influence clot size, permeability to fibrinolytic enzymes, and clot friability [127]. Moreover, appreciation of the role of red cell ligands, i.e., FasL-FasR, that mediate platelet-RBC interactions and facilitate VTE [128], suggest the potential discovery of novel mechanisms relevant to SCD.

## 4. Perspectives

A particularly vexing problem is that SCD mice do not spontaneously develop VTE, as patients with SCD do. Whether the absence of this phenotype simply reflects a failure to systematically document these occurrences or a fundamental difference between mice and humans is currently unknown. In addition, the mechanisms underlying the development of chronic venous vasculopathy, i.e., post-thrombotic syndrome, valvular venous insufficiency and leg ulceration in those experiencing DVT, and chronic thromboembolic pulmonary hypertension (CTEPH) in those experiencing PE are also unclear. The complex vascular pathobiological events occurring in SCD, particularly those concerning VTE, could be unraveled by developing murine models that recreate human disease pathophysiology with greater fidelity. Enhancing the efforts to develop additional SCD mouse models that are optimized to study DVT, PE and in-situ pulmonary thrombosis may therefore advance the field. A greater appreciation of these processes in SCD could also be gained by studying longitudinal cohorts of SCD patients experiencing thrombosis. Moreover, these studies could be augmented with investigations in animal models that are designed to provide mechanistic insights into the pathophysiology of chronic venous vasculopathy and permit manipulations that are impossible in human subjects. This synergy could lead to elucidation of the relative contributions of inflammatory and coagulation pathways and reveal novel mechanisms of VTE pathogenesis and venous vasculopathy. Thus, the murine models of SCD could advance our understanding of the occurrence of chronic complications of VTE, i.e., post-thrombotic syndrome and chronic thromboembolic pulmonary hypertension. 

Apparently, hemolysis driven inflammatory and coagulation processes, are insufficient to explain VTE risk in the entire spectrum of SCD patients. All SCD genotypes with variable HbS sickling severity share a similarly high risk of VTE, which is challenging to explain if sickling related intravascular hemolysis truly drives VTE pathophysiology. Less severe SCD genotypes with low grade sickling induced hemolysis, such as heterozygote for HbS (HbSC) disease and S/Beta^+^ thalassemia, experience frequent thromboembolic events; sickle cell trait individuals having an HbS fraction of ~35–40% exhibit a twofold higher risk of PE, compared with ethnic controls [13,116,129]. Therefore, additional studies are required to understand factors, such as higher hemoglobin and hitherto unappreciated genetic or environmental factors, that may explain this increased propensity for VTE. For instance, studies in SCD mice that are P-selectin deficient [120] could unravel the phenotypic heterogeneity of thrombosis in SCD induced by genetic variations in P-selectin. Moreover, environmental and age induced genetic variation may explain the development of a prothrombotic state [130,131]. Exploring these intriguing clinical observations could provide mechanistic insight into VTE and help to develop therapeutic solutions that would advance patient care and improve quality of life. Finally, in the context of COVID-19, elucidating the mechanisms of sepsis mediated coagulopathy has become highly relevant, since it predicts adverse outcomes [132]. Given the preexistent inflammation and coagulation dysregulation ongoing in SCD patients at baseline, it is possible that COVID-19 associated coagulopathy may worsen outcomes in these patients. Thus, studies of the effects of SARS-CoV2 infection in murine models of SCD might provide insight into the dysregulation of thromboinflammatory processes associated with these two diseases.

## 5. Conclusions

It is clear from human population studies, that coagulation factor or natural anticoagulant factor levels influence the risk of venous thrombosis, but it is equally clear that other factors are also contributory. An increased recognition that inflammatory diseases are associated with high risk for venous thrombosis suggests that innate inflammatory and coagulation responses are pathophysiologically relevant to this process. In addition, the complex multifactorial nature of venous thrombosis suggests that several risk factors, e.g., cancer, obesity, and a sedentary lifestyle, effect the likelihood of its development. Attempting to understand the complex genetic and environmental risk factors that influence VTE in complex inflammatory disorders could provide an opportunity to understand how these interrelated pathways initiate and propagate VTE. Dysregulated inflammation and overactive coagulation in SCD result in a triad of hypercoagulability, vessel wall dysfunction, and stasis. This unique prothrombotic milieu in SCD provides an opportunity to gain fundamental insights into the molecular and cellular aspects of VTE pathophysiology. Moreover, advancing our understanding of the mechanisms involved in chronic venous vasculopathy could lead to the development of new treatments and reduce VTE associated morbidity and mortality.

## Figures and Tables

**Figure 1 ijms-21-05279-f001:**
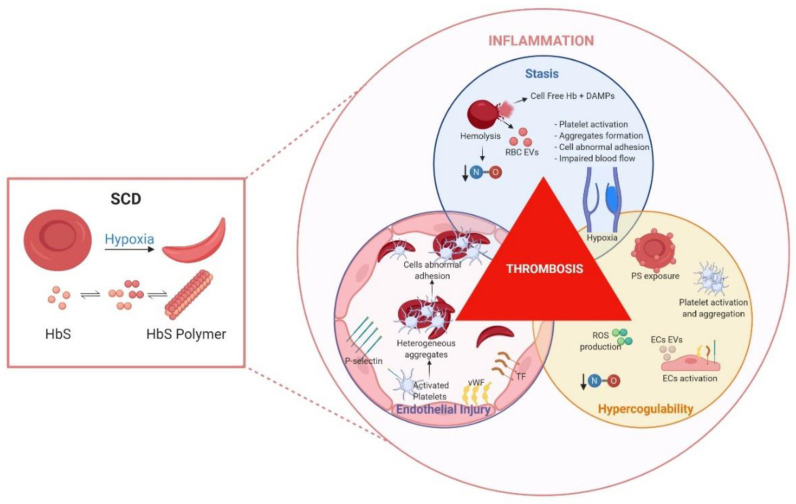
Sickle cell disease factors that contribute to thrombosis.

**Table 1 ijms-21-05279-t001:** Venous thromboembolism (VTE) mouse models.

Venous Thrombosis Murine Model	Characteristics	Disadvantages	Mechanisms of Thrombus Formation
Venous stasis but ligation induced injury	Complete and permanent occlusion of the inferior vena cava (IVC) and the venous flow.	The absence of blood flow, which not reproduce the clinical scenario where a thrombus is non-occlusive.	The combination of venous stasis and endothelial injury with upregulated expression of endothelial adhesion/procoagulant molecules imitates thrombosis.
Venous stenosis with no injury	Preservation but markedly reduced venous blood flow and minimal endothelial injury.	Occasional failure to induce persistent thrombosis and variability in thrombus size.	Endothelial activation, recruitment of immune cells and platelets, initiate thrombosis which is augmented by stasis.
Ferric chloride induced injury	Surgical exposure of IVC followed by topical application of ferric chloride.	Endothelial injury induced by the chemical irritant stimulus. Exposure time and concentration influence size and thrombus growth dynamics.	Oxidative damage to vascular endothelial cells.
Rose Bengal induced injury	This model is mainly used to induce acute thrombosis.	Endothelial injury by oxygen free radical induced oxidative stress.	Endothelial activation/injury with subsequent induce thrombus formation.
Electrolytic vein injury with local hypercoagulability	Non-occlusive thrombosis model that enables to study the acute and chronic deep vein thrombosis (DVT).	Substantial endothelial and vessel wall damage due to the needle access and lengthy procedural time.	Thrombus formation takes place after endothelial cell activation/injury, but blood flow is unaltered.

**Table 2 ijms-21-05279-t002:** Sickle cell disease (SCD) mouse models.

Mice Model	Type of Transgene	Phenotype	Limitations
**β^SAD^** **NY1DD**	Human α2-globin linked to a human β-globin LCR; human βSAD-globin gene carrying Antilles and Hb D-Punjab (β121Gln) variants linked to a human β-globin LCR	Mild phenotype Increased red cell density Low oxygen affinity and an enhanced polymerization potential Under hypoxia conditions, these mice express a more severe pathology Mice develop priapism, kidney defects, and shortened survival	Not anemic Mouse hemoglobin expression Genetic thalassemia background
**Aβ^S Antilles^** **S+SAntilles**	Human α2-globin and βS Antilles-globin variant (β23Ile), each linked to individual LCR HSII fragments	Moderate phenotype Anemic mice with low solubility and low oxygen affinity Slightly reduced hematocrit and haptoglobin levels Exhibit symptoms of VOC Increased reticulocyte count and plasma hemoglobin	Mouse hemoglobin expression Genetic thalassemia background
**Berkeley model** **SS-BERK**	Mini-LCR expressing human α1, Gγ, Aγ, δ, βS globins on a murine α- and β-globin-deficient background	Severe phenotype Express almost exclusively human sickle hemoglobin Sickle red blood cells (RBCs), intravascular hemolysis, reticulocytosis, severe anemia, leukocytosis, elevation of inflammatory cytokines, multiorgan infarcts, and pulmonary congestion Exhibit VOCs, I/R pathophysiology and increased inflammatory response Hyperalgesia	Low mean corpuscular hemoglobin concentrations (MCHC) Enlarged spleen with compensatory extramedullary hematopoiesis
**Townes model**	Human mini-LCR expressing human α1, Aγ, βS globins on a murine α- and β-globin-deficient background	Severe phenotype Expansion of red pulp, pooling of sinusoidal RBCs, vaso-occlusion, and loss of lymphoid follicular structure Marked reduction in RBC counts, Hb concentrations, PCVs, and a significantly increased reticulocyte count Hyperalgesia	Enlarged spleen with compensatory extramedullary hematopoiesis

I/R: Ischemia/Reperfusion; LCR: Locus control region; MCHC: Mean corpuscular hemoglobin concentrations; PCV: Packed cell volume, VOC: Vaso-occlusive crisis.

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
