# Peer review of "Sickle Cell Disease: A Paradigm for Venous Thrombosis Pathophysiology"

_ijms, 2020, doi:10.3390/ijms21155279_

Round 1
Reviewer 1 Report
In this review, Lizarralde-Iraggori and Shet comprehensively discuss the literature about the procoagulant milieu of sickle cell disease in an effort to understand the increased incidence of VTE in this patient population compared to ethnic- and age-matched healthy controls. I like the way that the data was discussed and organized as the three parts of Virchow's Triangle, it was an interesting and novel way of organizing this type of review. This review is very comprehensive, yet well-written and engaging.
Major Comment
- In several places, the authors cite review articles rather than primary literature. In particular line 67 (refs 16,17), line 70 (refs 15, 18), line 135 (refs 25-27), line 255 (citation 63), line 285 (ref 70), line 320 (ref 18), would benefit from referencing the original studies so that readers can access those data directly.
- The authors discuss the increased incidence of VTE in SCD patients compared to healthy controls. In sickle cell disease, there is an imbalance that favors pulmonary embolism over deep vein thrombosis, but this was not broadly discussed in the review. A brief summary of this paradox should be included in this review.
Minor Comment
- Please check the labeling of headings/subheadings. Section 2 is titled "Animal models of disease pathophysiology" but the subheadings were 3.1 and 3.2, etc.
Author Response
July 22nd, 2020
Re: IJMS-863987
Dear IJMS Editors & Reviewers:
Thank you for reviewing our manuscript. We appreciate the suggestions of both reviewers in response to our manuscript. Please find below a point-by-point response to each of these suggestions. We hope that the resulting modifications to the manuscript are suitable and thank you for your time and consideration.
Sincerely Yours,
Maria A. Lizarralde-Iragorri
Reviewer #1
Major Comment
- In several places, the authors cite review articles rather than primary literature. In particular line 67 (refs 16,17), line 70 (refs 15, 18), line 135 (refs 25-27), line 255 (citation 63), line 285 (ref 70), line 320 (ref 18), would benefit from referencing the original studies so that readers can access those data directly.
- Thank you very much for this observation. We have now cited the primary literature.
- The authors discuss the increased incidence of VTE in SCD patients compared to healthy controls. In sickle cell disease, there is an imbalance that favors pulmonary embolism over deep vein thrombosis, but this was not broadly discussed in the review. A brief summary of this paradox should be included in this review.Thank you for this comment.
- We now include this in the section introducing SCD (see page 2, line 55-59).
Minor Comment
- - Please check the labeling of headings/subheadings. Section 2 is titled "Animal models of disease pathophysiology" but the subheadings were 3.1 and 3.2, etc.
- The numbering of the headings has been corrected.

Reviewer 2 Report
The review by Lizarralde-Iragorri and shet addresses deep vein thrombosis in the context of sickle cells disease. The review is well written and very comprehensive. I only have minor comments:
- The authors discuss the indirect effect of DAMPs on platelet activation further potentiating thrombosis. However many of these damps, in particular haemoglobin and heme, were shown to directly activate platelets through GPIb and CLEC-2. Moreover, other damps released in SCD such as histones can also activate platelets through TLR2/TLR4. A direct contribution of DAMPs to platelet activation can be added to the review, independently of indirect effects.
- Although minor roles for platelets were described in DVT, however recent evidences have shown an important role for platelet receptors in DVT. This aspect can also be discussed in the review.
Author Response
July 22nd, 2020
Re: IJMS-863987
Dear IJMS Editors & Reviewers:
Thank you for reviewing our manuscript. We appreciate the suggestions of both reviewers in response to our manuscript. Please find below a point-by-point response to each of these suggestions. We hope that the resulting modifications to the manuscript are suitable and thank you for your time and consideration.
Sincerely Yours,
Maria A. Lizarralde-Iragorri
Reviewer #2
The review by Lizarralde-Iragorri and Shet addresses deep vein thrombosis in the context of sickle cells disease. The review is well written and very comprehensive. I only have minor comments:
- The authors discuss the indirect effect of DAMPs on platelet activation further potentiating thrombosis. However many of these damps, in particular haemoglobin and heme, were shown to directly activate platelets through GPIb and CLEC-2. Moreover, other damps released in SCD such as histones can also activate platelets through TLR2/TLR4. A direct contribution of DAMPs to platelet activation can be added to the review, independently of indirect effects.
- Thank you for this suggestion. We now reference the contribution of DAMPs on platelet activation by describing the impact of the histones in coagulation and thrombin generation through TLR2/TLR4. (page 6, lines 340-3342)
- Although minor roles for platelets were described in DVT, however recent evidences have shown an important role for platelet receptors in DVT. This aspect can also be discussed in the review.
- Thank you for this comment. We agree that there is potential to evaluate a role for CLEC-2 in SCD and we have included this in the revised manuscript (page 6, lines 358-361).